# Shape of the first mitotic spindles impacts multinucleation in human embryos

Yuki Ono [1] ✉, Hiromitsu Shirasawa[1], Kazumasa Takahashi [1], Mayumi Goto[1], Takahiro Ono [2], Taichi Sakaguchi[1], Motonari Okabe[1], Takeo Hirakawa [1], Takuya Iwasawa[1], Akiko Fujishima [1], Tae Sugawara[1], Kenichi Makino[1], Hiroshi Miura[1], Noritaka Fukunaga [3], Yoshimasa Asada [3], Yukiyo Kumazawa[1] & Yukihiro Terada [1]

During human embryonic development, early cleavage-stage embryos are more susceptible to errors. Studies have shown that many problems occur during the first mitosis, such as direct cleavage, chromosome segregation errors, and multinucleation. However, the mechanisms whereby these errors occur during the first mitosis in human embryos remain unknown. To clarify this aspect, in the present study, we image discarded living human two-pronuclear stage zygotes using fluorescent labeling and confocal microscopy without microinjection of DNA or mRNA and investigate the association between spindle shape and nuclear abnormality during the first mitosis. We observe that the first mitotic spindles vary, and low-aspect-ratio-shaped spindles tend to lead to the formation of multiple nuclei at the 2-cell stage. Moreover, we observe defocusing poles in many of the first mitotic spindles, which are strongly associated with multinucleation. Additionally, we show that differences in the positions of the centrosomes cause spindle abnormality in the first mitosis. Furthermore, many multinuclei are modified to form mononuclei after the second mitosis because the occurrence of pole defocusing is firmly reduced. Our study will contribute markedly to research on the occurrence of mitotic errors during the early cleavage of human embryos.

During human embryonic development, many errors, such as chromosome mis-segregation, could occur that lead to aneuploidy[1–3], which originates during early mitosis[1,4–6]. The first mitosis is different from the subsequent mitosis because it includes the fusion of paternal and maternal genomes. The parental genome separates each pronuclear envelope, and the separated genome is fused and arrayed in the spindle after pronuclear envelope breakdown. An accurate first mitosis is essential to prevent genetic defects; however, studies have shown that several errors could occur during the first mitosis, such as direct cleavage, chromosome segregation errors, and multinucleation[6–8]. A live-cell imaging study using time-lapse confocal microscopy showed that the frequency of multinucleation at the 2-cell stage was 78% in human embryos[9]. Although multinucleated embryos transferring could lead to birth of babies, embryos with multiple nuclei at the 2-cell stage have lower blastocyst developmental rate and live birth rates than those with normal nuclei[10]; thus, nuclear errors can be used as a deselection biomarker for embryo transfer[9,11–14].

Many studies have revealed the abnormalities of DNA and cytoskeleton in the early cleavage stage of human embryos using fixed and immunofluorescence methods. Spindle abnormalities, such as multipolar spindles, are often observed in human zygotes[15,16]. Observation

[1]Department of Obstetrics and Gynecology, Akita University Graduate School of Medicine, Akita, Japan. [2]Department of Neurosurgery, Akita University Graduate School of Medicine, Akita, Japan. [3]Asada Institute for Reproductive Medicine, Asada Ladies Clinic, Nagoya, Aichi, Japan. ✉e-mail: onoy@med.akita-u.ac.jp

of spindle and nuclear abnormalities on days 3 and 5 in fixed human embryos showed that the frequency of abnormalities was higher in cleavage-stage embryos than in blastocyst-stage embryos[17]. However, the mechanism by which nuclear and spindle abnormalities occur remains largely unknown. Therefore, live-cell imaging of early cleavage-stage embryos is required for further investigation.

Microinjections of DNA or mRNA into mammalian embryos were widely used for live imaging during early mitosis[18–22]. Mouse embryos have dual spindles during the first mitosis, and the parental chromosomes remain apart until the second mitosis[23]. However, there are differences between mouse and human zygotes; therefore, the results obtained from studies on mouse zygotes do not always agree with those of studies on human zygotes. For example, only 5% of mouse embryos have abnormal nuclei at the first cell division[24], and centrioles of sperm are not inherited because sperm centrioles degenerate during spermiogenesis[25]. In mouse zygotes, acentriolar cytoplasmic microtubule-organizing centers are formed during the first divisions[23,26]. Moreover, microinjections of DNA or mRNA into human embryos pose the risk of ethical issues[27]. In addition, most human embryos used in research are donated surplus embryos by patients treated at fertility clinics. Surplus human embryos are mostly blastocysts, and it is difficult to obtain a sufficient number of human zygotes. Therefore, these factors hinder the progress of cell biology research on early human embryonic development.

Some recent studies show live imaging of human zygotes. These studies showed that chromosome mis-segregation often occurs during the first mitosis stage in human embryos[16] and inappropriate pronuclei migration and chromosome clustering at the interface can cause lagging chromosomes leading to chromosome loss[22]. Moreover, a study using live imaging without genetic manipulation for human blastocysts showed that blastocyst mechanical stress from blastocyst expansion or biopsy triggers nuclear DNA loss[28]. These studies on live-cell imaging of human embryos can reveal the mechanisms by which errors frequently occur in human embryos.

The present study focused on the movement of chromosomes and spindles during the first mitosis; we aimed to establish a method used for real-time imaging of DNA and microtubules in human two-pronuclear stage zygotes using chemical fluorescent labeling. We discovered that spindle shape was associated with blastomere nucleation status, spindle pole defocusing caused multinucleation, and differences in the positions of centrosomes cause spindle abnormality during the first mitosis in human embryos. Furthermore, we observed that many multinuclei are modified to form mononuclei after the second mitosis because pole defocusing was firmly reduced. Our study contributes to the identification of the causes of nuclear errors and developmental processes in human embryos.

## Results

### Live-cell imaging during the first mitosis in human embryos

We imaged human zygotes to observe the dynamics of DNA and microtubules during first mitosis. Figure 1a and Supplementary Movie 1 show representative confocal microscopy images of human embryos using fluorescent labeling. SPY-505DNA visualized DNA in the parental pronuclei, and DNA condensed into chromosomes before pronuclear envelope breakdown (PNBD) (Fig. 1a, SPY505-DNA (i)). Before the chromosomes were aligned, the spindle poles were visualized using SPY650-Tubulin (Fig. 1a, SPY650-Tubulin (ii)). In all imaged zygotes, the sperm tail was attached to the spindle poles (Fig. 1a, SPY650-Tubulin (iii), white arrowhead). When chromosomes segregate, the spindle elongates and forms the midbody (Fig. 1a, SPY650-Tubulin (iv), (v), Merge (vi)). Many granules labeled with SPY-505DNA and SPY-650 tubulin were imaged in all embryos and were observed under bright field (Fig. 1a, Bright field, white arrow). Cytoplasmic granules are slightly ovoid organelles formed in oocytes and zygotes[29,30], and a previous study showed that human

embryos store proteins such as chromatin modification factor and tubulin in cytoplasmic lattices[31]. Figure 1b shows multipolar segregation (Supplementary Movie 2). A multipolar spindle was formed from three poles (Fig. 1b (ii) white arrow). The chromosomes did not array linearly but had a 'Y' shape and were segregated in multipolar (Fig. 1b (iii), (iv)).

To ensure that this method did not inhibit the dynamics of DNA and tubulin during the first mitosis, we compared the time from PNBD to furrow ingression between embryos of a control, an imaged, and a dyed group (Fig. 1c, Supplementary Fig. 1). Embryos in a control group were incubated in medium not containing chemical dye like SPY-505DNA and SPY-650tubulin; those in an imaged group were incubated in those with dye and imaged using a confocal microscopy; and those in a dyed group were incubated in medium with SPY dyes but not imaged. The mean times for the control, imaged, and dyed groups were 149.2, 155.6, and 155.1 min, respectively. No significant differences were observed between the groups, suggesting that visualizing chromosomes and tubulin using our method does not significantly disturb the progression of the first mitosis in human embryos.

Figure 1d shows the frequency of anaphase errors during the first mitosis. We observed five embryos undergoing multipolar segregation, and only one of them showed direct cleavage, a type of abnormal cleavage wherein one embryo divides into three or more blastomeres (Fig. 1d, second bar).

Figure 1e shows the nuclear status at the 2-cell stages. The mononuclear type has one large nucleus, and if there are other nuclei, their diameters are usually smaller than 10 μm. The multinuclear type does not have one large nucleus but has nuclei of various sizes in either or both of the cells in the 2-cell stage. Figure 1f shows the frequency of the nuclear status at the 2-cell stage. The frequency of the multinuclear type was higher than that of the mononuclear type (81% ($n = 25/31$), 19% ($n = 6/31$)). All embryos undergoing multipolar segregation demonstrated a multinuclear status (Fig. 1f, third bar). The rate of multinucleation for the control and dyed groups were 75% (15/20) and 78% (25/32), respectively (Fig. 1g). No significant differences were observed between the groups; the frequency of the multinuclear type was 81% ($n = 25/31$) in an imaged group, suggesting that the chemical dye and laser scanning used in our method does not significantly cause multinucleation at the two-cell stage in human embryos. Therefore, these suggests that anaphase and nuclear errors often occur in 2-cell stage human embryos.

### Spindle shapes are firmly associated with multinucleation

Various types of spindles were observed in this study (Supplementary Figs. 2–4). We analyzed spindles at the end of metaphase from an angle where the chromosomes appear to be aligned linearly and where the spindle appears to have the maximum area and calculated the aspect ratio of the spindles (Fig. 2a, b (i)–(iii)). Figures 1a and 2a show representative images of high- and low-aspect-ratio (AR) spindles, respectively. The high-AR spindle was sharp, elongated, had focused poles, and led to a mononuclear type at the 2-cell stage (Fig. 2a (iv)–(vi), Supplementary Movie 3). The low-AR spindle appeared as dual spindles with defocused poles and chromosomes that could not unify during anaphase and became multinucleated (Fig. 2b (iv)–(vi), Supplementary Movie 4). Figure 2c showed that there was a significant difference between the AR of the spindles leading to the mononuclear and multinuclear types at the 2-cell stage. The results, including only bipolar segregation, did not change (Fig. 2d). Figure 2e shows the AR of all imaged spindles ($n = 31$). These results suggest that nuclear formation in 2-cell-stage embryos is strongly associated with spindle shape.

### Spindle pole defocusing causes multinucleation

Figure 3a shows a spindle with unilaterally defocused poles (Supplementary Movie 5). This spindle started to form from two centers of

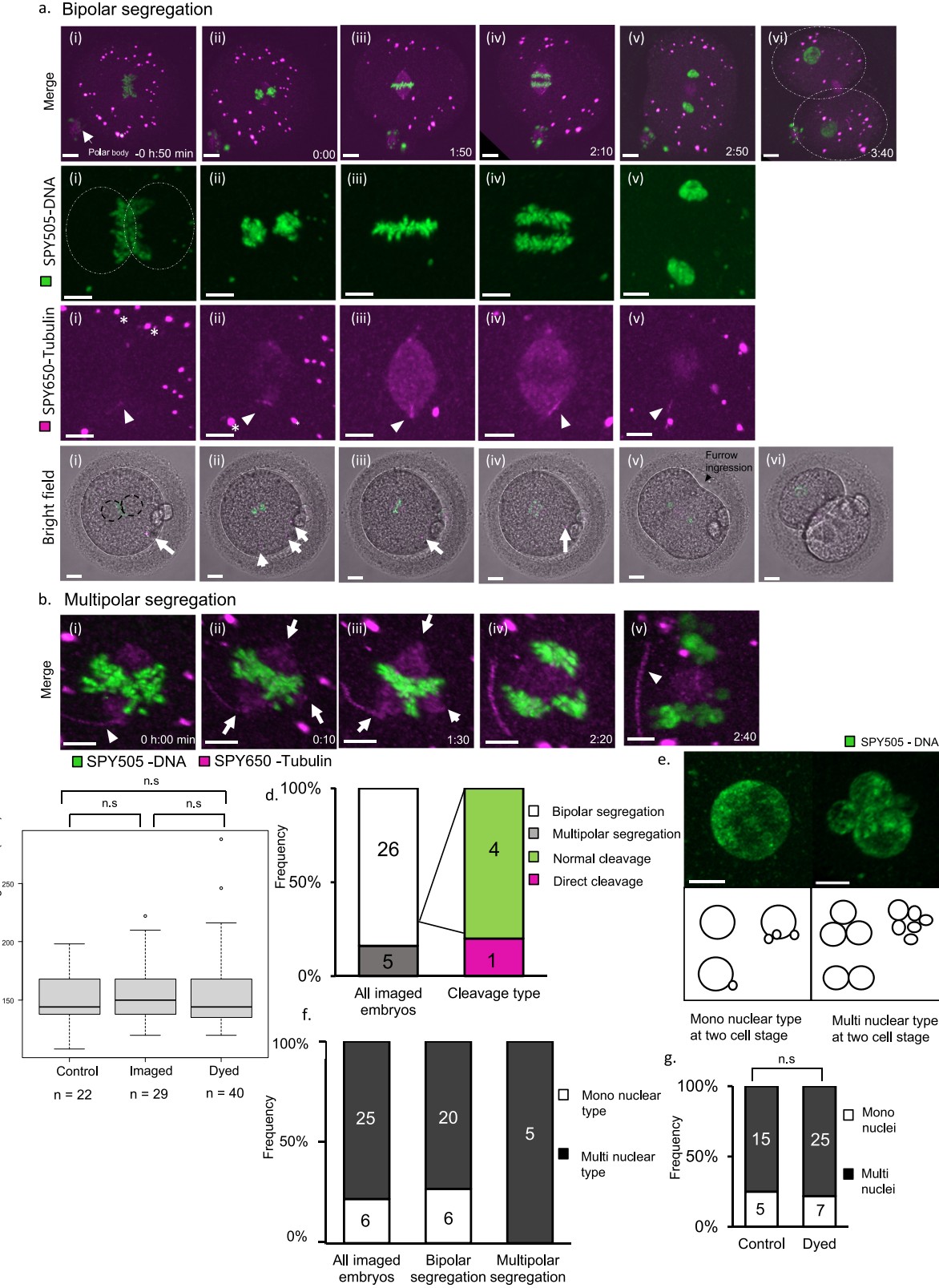

cytoplasmic microtubule nucleation and formed two polar spindles (Fig. 3a (i) and (ii)), but the unilateral pole became defocused in the last half of metaphase (Fig. 3a (iii), white arrow), leading to the development of a multinuclear-type 2-cell embryo. Figure 3b shows a spindle with bilateral defocused poles (Supplementary Movie 6). This spindle also started to form from two spindle poles; a sperm tail was attached at the center of the pole (Fig. 3b (i) and (ii), white arrow), but the

bilateral pole became defocused, and the sperm tail moved to the edge of the defocused pole in the last half of metaphase (Fig. 3b (iii), white arrow), making it a multinuclear-type 2-cell embryo. Figure 3c shows the significant difference in the AR of the spindles between focusing and defocusing poles in an embryo undergoing bipolar segregation. We often observed polar defocusing in the first mitotic spindle during metaphase (42%, $n = 11/26$) in the embryo undergoing bipolar

**Fig. 1 | Live-cell imaging using fluorescent labeling during the first mitosis in human embryos. a**, Time-lapse live imaging of representative confocal microscopic images of human embryos undergoing bipolar segregation using fluorescent labeling and images captured in a bright field simultaneously. Time in hours:min, 0:00 is time when Pronuclear envelope breakdown occurs. Merge: scale bar, 15 μm; SPY505-DNA, SPY650-Tubulin: scale bar, 10 μm; bright field: scale bar, 20 μm. White arrowheads indicate the sperm tail. The white asterisks indicate cytoplasmic granules labeled with SPY505-DNA and SPY650-Tubulin. **b** Time-lapse live imaging of representative confocal microscopy images of human embryos undergoing multipolar segregation. White arrowheads indicate the sperm tail. The three white arrows indicate the poles of the multipolar spindles. Note that the three poles formed in early metaphase and did not change until anaphase. Scale bar, 10 μm. **c** Quantification of the timing from PNBD to furrow ingression for embryos in the control, imaged, and dyed groups. Embryos in the dyed group were not imaged and were stained with the SPY dyes in a time-lapse incubator. Box and whisker plots represent minimum, lower quartile, median, upper quartile, and maximum. Outliers fall beyond these lines. The bottom of the box is the first quartile (25th percentile) and the top of the box is the third quartile (75th percentile). The line in the middle of the box is the median (50th percentile). The Kruskal-Wallis test was performed, *p*-value = 0.5044. Source data are provided as a Source Data file. **d** Quantification of anaphase errors in all imaged embryos. The number of embryos in which cleavage errors occurred during multipolar segregation is shown in the second bar. The numbers are shown as bars. Source data are provided as a Source Data file. **e** The nuclear status at the 2- and 4-cell stages. The mononuclear type has one large nucleus, and if other nuclei were present, their diameters were <10 μm. The multinuclear type does not have one large nucleus but nuclei of various sizes in either or both of the cells in the 2-cell stage. Scale bar, 10 μm. **f** Quantification of nuclear errors in all imaged embryos. The second bar shows the number of embryos undergoing only bipolar segregation, and the third bar shows only multipolar segregation. **g** Quantifying nuclear errors in embryos in a control group and a dyed group. *P*-value from two-sided Fisher's exact test. p-value = 1. Source data are provided as a Source data file.

segregation (Fig. 3d). Moreover, the frequency of defocused poles was significantly lower in mononuclear-type embryos than in multipolar-type embryos (0% (*n* = 0/6), 55% (*n* = 11/20, *P* < 0.05)) (Fig. 3d). Figure 3e shows the frequency of the spindles with defocused poles in an embryo undergoing bipolar segregation. Among 11 spindles with defocused poles, the spindles with unilateral defocused poles and bilateral defocused poles were observed at similar frequencies (45% (*n* = 6/11), 54% (*n* = 5/11)), and the defocused pole side was more often the pole not attached to the sperm tail (*n* = 4/6). Subsequently, we investigated spindles leading multinuclear type and having focused poles, not defocused poles (Fig. 3d, first bar, *n* = 9). Figure 3f shows that even excluding embryos that exhibited defocused poles, there was a significant difference between the AR of the spindle leading to mononucleated and multinucleated types. These results indicated that low AR spindles often induce multinucleation regardless of whether they have focusing or defocusing poles. In addition, poles of high-AR spindles rarely become defocused, whereas those of low-AR spindles often become those with defocused poles., and this leads to multinucleation at the 2-cell stage.

## Multipolar spindles and defocused spindles differ
Low-AR spindles including defocused spindles and multipolar spindles cause multinucleation at the two-cell stage (Figs. 1–3). Both spindles are similar in that they do not have two focused poles. We investigated the formation process to examine their differences. Spindle formation began 10 min (IQR = 10 min) before PNBD and it took 30 min (IQR = 10.8 min) for the embryo to complete spindle formation. Moreover, the duration of spindle elongation from PNBD was 130 min (IQR = 11.0 min). A blue dashed line in Fig. 4a indicates the onset of spindle poles defocusing. Spindles with defocused poles were formed from two poles and became bipolar spindles with two focused poles during the first half of metaphase. The onset of spindle pole defocusing occurred 80 min (IQR = 19.0 min) from PNBD. Spindle morphology changed from two poles to defocused poles in the last half of the metaphase. Meanwhile, multipolar spindles with multipolar segregation were formed from three poles and appeared much earlier than those with defocused poles (Fig. 4a, black dashed line at 15.0 min (IQR = 12.5 min) vs 94.5 mins (IQR = 32.75 min), from the beginning of spindle formation). There were no significant differences in the duration of phases in the first stage of mitosis between the spindle types and nuclear type (Tables S1–2). Next, we investigated the position of centrosomes in different morphologic spindles using immunofluorescence. We used γ-tubulin to determine where centrosomes exist in spindle poles. γ-tubulin is an essential centrosome protein that is required for the formation of nucleation sites at MTOCs[32,33]. We imaged human zygotes using the same method, and fixed embryo approximately one hour after the onset of PNBD. High-aspect ratio spindle with focused poles had two γ-tubulin positive poles, and the two poles lined up on a line perpendicular to the chromosomes (Fig. 4b (i)). On the other hand, spindles with defocused poles had two γ-tubulin positive poles, but the position was offset from the line perpendicular to the chromosomes (Fig. 4b (ii)). Multipolar spindles appeared to have three poles, but actually only have two γ-tubulin positive poles and the other pole was a γ-tubulin negative pole (Fig. 4b (iii)). Therefore, spindles with defocused poles are significantly different from multipolar spindles owing to the formation process and the location of the centrosome. These results show that multipolar spindles and defocused spindles causing multinucleation have a different formation process and the position of centrosomes are different.

## Stable second mitotic spindles modify multinuclei
Next, we compared the frequency of the nuclear status between the 2- and 4-cell stages. To reduce phototoxicity to embryos, we limited the duration of confocal microscopic observation to 8.4 ± 3.1 [standard deviation (SD)] h. We imaged the embryos consecutively during the first mitosis, after which the imaged embryos were moved into a time-lapse incubator. After the second division, the embryos were imaged using a confocal microscope (Supplementary Fig. 1). The frequency of the multinuclear type was 81% and 35% at the 2- and 4-cell stages, respectively (Fig. 5a).

We measured the diameter of nuclei in each blastomere in the 2- and 4-cell stage embryos (Fig. 5b) and investigated how the nuclear diameter changed after the second mitoses. Figure 5c, d shows the same embryos and the nuclear diameters of all embryos that were imaged during the first mitosis and after the second mitosis (*n* = 11). There was a wide range of variations in plots of the diameter at the 2-cell stages, while nuclear diameters of approximately 20 μm were highly frequent at the 4-cell stage. The pink bar under 10 μm shows micronuclei caused by a lagging chromosome during the first mitosis. The lagging chromosome occurring during anaphase did not merge with the mono nucleus becoming a micro nucleus (Fig. S5a, Supplementary movies 7 and 8). There were no significant differences in spindle pole status, nuclear type, and spindle AR whether the embryos had lagging chromosomes or not (Figure S5b–d). Therefore, it is suggested that lagging chromosomes occur independent of spindle shape.

Figure 5c, d shows that medium-sized nuclei (7–18 μm) of the 2-cell stage mostly disappeared and small-sized nuclei (under 10 μm) were left at the 4-cell stage. To investigate how the phenomenon occur, we also imaged human embryos during the second mitosis (*n* = 9). Figure 5e–g shows how often the second mitotic missegregation and spindles defocusing occurs. Interestingly, no spindle pole defocusing was observed in the second mitosis and the occurrence of spindle poles defocusing during the second mitosis was significantly lower than those during the first mitosis (Fig. 5f, 42% (11/26)

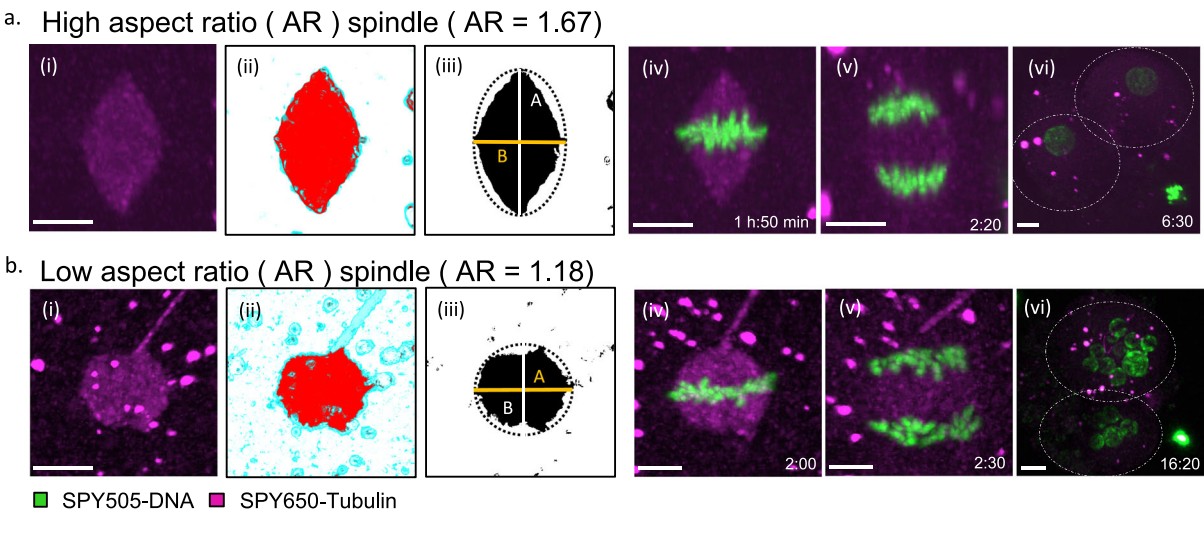

a. High aspect ratio ( AR ) spindle ( AR = 1.67)

b. Low aspect ratio ( AR ) spindle ( AR = 1.18)

■ SPY505-DNA ■ SPY650-Tubulin

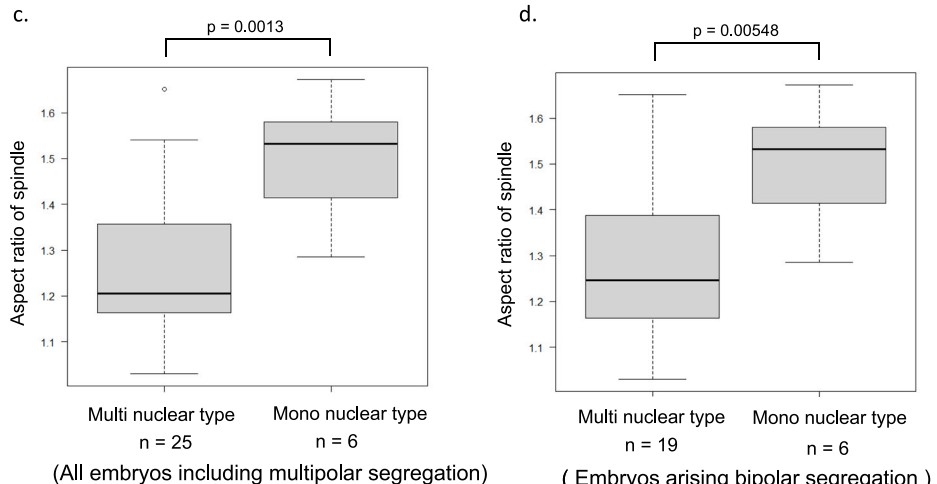

c.

p = 0.0013

Aspect ratio of spindle

Multi nuclear type
n = 25

Mono nuclear type
n = 6

(All embryos including multipolar segregation)

d.

p = 0.00548

Aspect ratio of spindle

Multi nuclear type
n = 19

Mono nuclear type
n = 6

( Embryos arising bipolar segregation )

e.

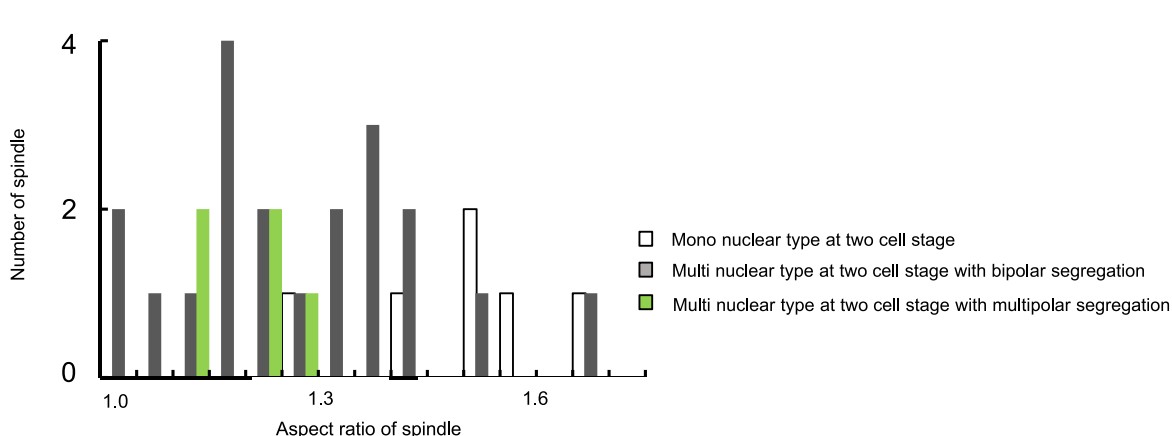

Number of spindle

Aspect ratio of spindle

□ Mono nuclear type at two cell stage
■ Multi nuclear type at two cell stage with bipolar segregation
■ Multi nuclear type at two cell stage with multipolar segregation

vs. 0% ($n = 0/7$)). The occurrence of multipolar segregation and a lagging chromosome was not significantly different between the first mitosis and the second mitosis (Fig. 5e, 16% ($n = 5/31$) vs. 22% ($n = 2/9$), Fig. 5g, 16% (5/31) vs 19% (1/9)). Figure 5h shows how the multinuclear type at the two-cell stage becomes mononuclear during the second mitosis (Supplementary Movie 9). The chromatin of each nucleus in the multinucleated cell condensed and formed chromosomes (Fig. 5h (iii)), and the chromosomes gathered into a second mitotic spindle (Fig. 5h (v)) and segregated into two separate areas (Fig. 5h (viii)). These results suggest that the second mitotic spindles were stable, and most of the multinuclei gathered and arrayed in the spindle, leading to the formation of mononuclei.

**Fig. 2 | Spindle shapes are firmly associated with multinucleation. a, b** (i) Representative high-aspect-ratio (AR) spindle (**a**) and low-AR spindle (**b**) images at the end of metaphase from an angle that looks like chromosomes aligning lineally and where the spindle appears to have the maximum area. (ii) Images processed using Ilastik. The objects were classified using an image recognition algorithm, and the shape of the spindle was extracted as a red area. The background and parts of the sperm tail were excluded, as indicated in blue. (iii) The binarized image of (ii) is created by extracting only the red area. AR, which is the ratio of the major(A) and minor axes (B), was calculated automatically using Fiji. (iv) Time-lapse live imaging of representative confocal microscopic images of human embryos exhibiting a high-AR (1.67) spindle (**a**) and low-AR (1.18) spindle (**b**). Scale bar, 10 μm. Scale bar for the picture at the right end is 15 μm. **c** Quantification of spindle AR for all image embryos of multinuclear and mononuclear types at the 2-cell stage. An unpaired

two-tailed Student's $t$-test was performed. Box and whisker plots represent minimum, lower quartile, median, upper quartile, and maximum. Outliers fall beyond these lines. The bottom of the box is the first quartile (25th percentile) and the top of the box is the third quartile (75th percentile). The line in the middle of the box is the median (50th percentile). Source data are provided as a Source Data file. **d** Quantification of spindle AR for imaged embryos undergoing bipolar segregation in multinuclear and mononuclear types at the 2-cell stage. Box and whisker plots represent minimum, lower quartile, median, upper quartile, and maximum. The bottom of the box is the first quartile (25th percentile) and the top of the box is the third quartile (75th percentile). The line in the middle of the box is the median (50th percentile). An unpaired two-tailed Student's $t$-test was performed. Source data are provided as a Source data file. **e** Histogram showing the AR distribution of all spindles grouped into 0.05 bins. Source data are provided as a Source data file.

## Discussion

The purpose of the present study was to image live human embryos using fluorescent labeling, without using the conventional microinjection strategy, for studying the mechanisms underlying the occurrence of nuclear abnormalities during the first mitosis. We used the fluorescent labels SPY505-DNA and SPY650-Tubulin and established a method to image the dynamics of DNA and microtubules during first mitosis. We demonstrated that there was variation in the first mitotic spindles and that low-AR spindles were much more likely to lead to multinucleation at 2-cell stage (Fig. 2c, d). In addition, the low-AR spindles were unstable and often had defocused poles, leading to multinucleation at the 2-cell stage (Fig. 3c, d). Moreover, we also discovered that there was difference in centrosome position on the poles between each spindles (Fig. 4b). Figure 6 shows the model of spindle instability causing multinucleation. In addition, we discovered the reduction in frequency of the multinuclear type was 81% and 35% from the 2-cell to 4-cell stages (Fig. 5a). The occurrence of spindle poles defocusing during the second mitosis was significantly lower than those during the first mitosis (Fig. 5f, 42% (11/26) vs. 0% ($n = 0/7$)), which may contribute to the stability of the second spindle (Fig. 5).

Our study showed that the first mitotic spindles varied and were strongly associated with multinucleation. Spindle instability rarely occurs in normal somatic cells[34]. Conventionally, mouse embryos are used as a substitute for human embryos to study the dynamics of DNA and the cytoskeleton during the cleavage stage. However, 80% of the first mitotic spindles in mouse embryos are barrel-shaped[35] and rarely lead to nuclear errors at the 2-cell stage (<5%)[24]. In bovine embryos, which show over 60% multinucleation at the 2-cell stage, a study showed that the first mitotic spindles that were formed after juxtaposed pronuclear envelope breakdown were stable, and over 80% of the spindles were normal spindle[36]. Spindle instability similar to the abnormal spindles in our study was observed in a study on human oocytes[37,38], and this meiotic spindle instability caused segregation errors in meiosis and led to aneuploidy in human eggs. The same study identified a molecular motor KIFC1 (kinesin superfamily protein C1) as a spindle-stabilizing protein; unlike mouse and bovine oocytes, human oocytes lack this protein[38]. Interestingly, Human zygotes exhibit low levels of KIFC1 mRNA[38]. It is possible that the first mitotic spindle instability in human embryos is inherited from oocytes.

In our study, we showed that there are two kinds of spindle that induce multinucleation at the two-cell stage: low-AR spindles which often have defocused poles and multipolar spindles. They were divided depend on the formation time. Although low-AR spindles with defocused poles were formed from two poles and became bipolar spindles with two focused poles during the first half of metaphase, the spindle morphology changed from two poles to defocused poles in the last half of metaphase. Meanwhile, multipolar spindles with multipolar segregation were formed from three poles and appeared much earlier than those with defocused poles (Fig. 4a,

black dashed line, 15.0 min (IQR = 12.5 min) vs 94.5 min (IQR = 32.75 min), from the beginning of spindle formation). We also observed the sperm tail moving toward the edge of the defocused pole from the center during the last half of metaphase (Fig. 3b(ii), (iii) blue arrowhead, Supplementary Movie 6). A previous study showed that a sperm had a typical proximal centriole and atypical distal centriole, and the distal centriole exists at the edge of the sperm tail and is inherited by zygotes after fertilization[39]. Furthermore, sperm centrioles devoid of pericentriolar matrix (PCM) recruit maternal PCM proteins upon reaching the egg during fertilization, forming a functional centrosome for embryo development In most species[40]. In bovine zygotes, the centrosomes did not function as centers of cytoplasmic microtubule nucleation but are only loosely connected to spindles[36]. However, it remains unknown how inherited sperm centrosome and maternal PCM form centrosomes during the first mitosis in humans. It is possible that the movements of the defocused poles and sperm tail indicate that the human zygote tends to have dysfunction of the centrosome inherited from the sperm during the first mitosis.

Moreover, the two kinds of spindles had different positions of γ-tubulin positive MTOCs, respectively. Multipolar spindles had three spindle poles but had only two γ-tubulin positive poles and the other pole was γ-tubulin negative (Fig. 4b(iii)). This result means that human zygotes can make MTOCs without centrosomes inherited from sperm like human oocytes. Meanwhile, spindles with defocused poles had two γ-tubulin positive poles, but the position was offset from the line perpendicular to the chromosomes (Fig. 4b(ii)). The long term of mitotic division may cause defocusing poles, because pole defocusing occurs during the last half of metaphase. The duration from PNBD to anaphase during the first mitosis is significantly longer than the second mitosis in human embryos[16]. Therefore, the centrosomes consisted of inherited sperm centriole, and maternal PCM may not be functionally maintained until the anaphase as MTOC. In fact, there were significantly reduced rates of poles defocusing from the first to second mitosis in our study. In other words, spindle stability may change as cell division progresses. Further investigation is necessary to determine whether improvements in spindle instability occur after the second mitosis.

This study has several limitations. First, human embryos were donated from a fertility clinic, and detailed patient information, such as underlying disease and method of ovarian stimulation, was not available. It is possible that the human zygotes used in this study had a bias in patient characteristics. Second, although we reported several phenomena like varying first mitotic spindles and defocused poles, we could not uncover all the underlying mechanism. Our study suggested that spindles with poles defocusing and multipolar spindles had different positions of γ-tubulin positive MTOCs compared to high-AR spindles. However, we did not reveal why low-AR spindles with focused poles tend to form multinuclei compared to high-AR spindles. In our study, four γ-tubulin spots during anaphase were observed (Fig. S6, whiteheads). This suggested that zygotic

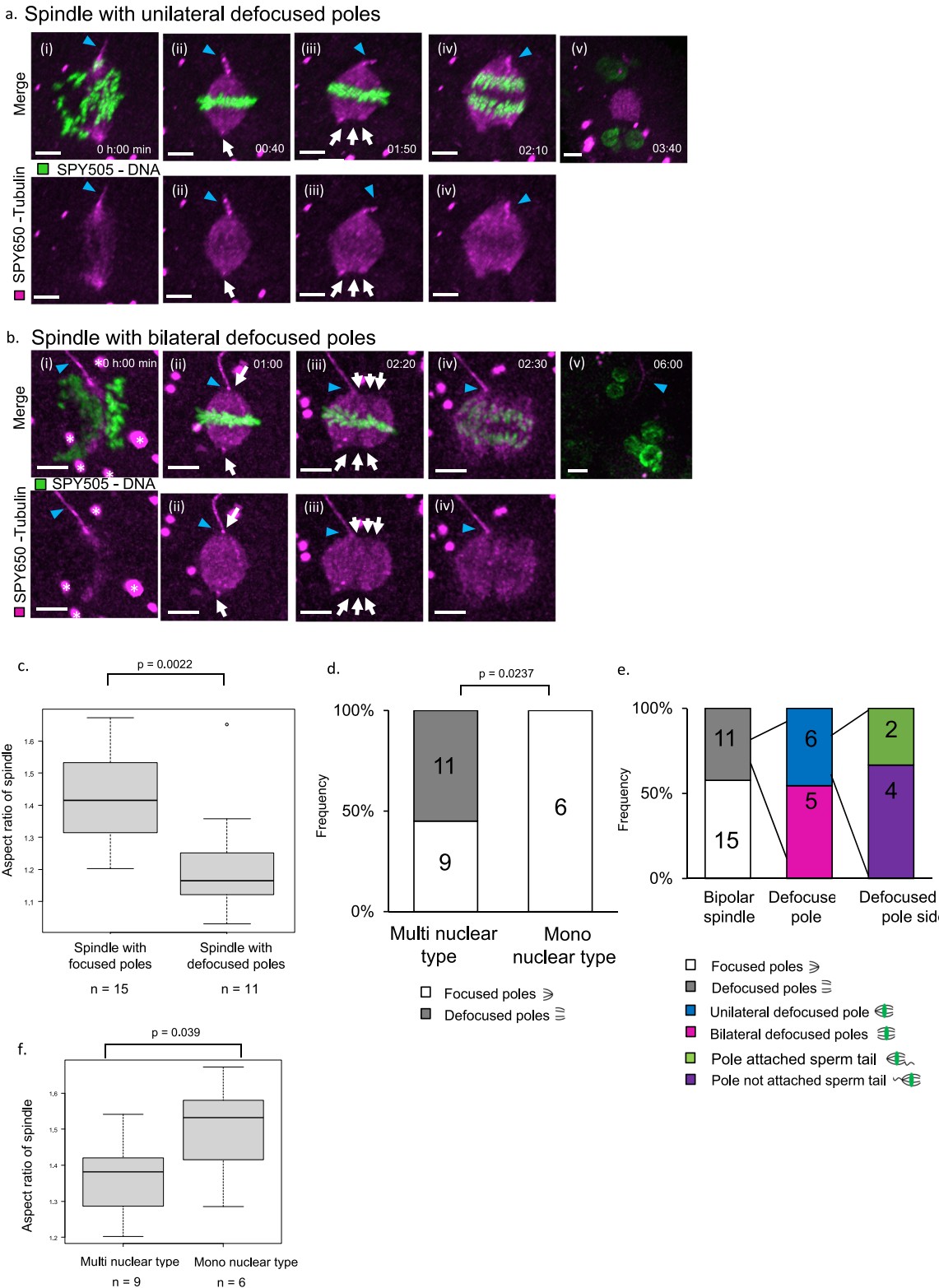

c. p = 0.0022

d. p = 0.0237

e.

f. p = 0.039

Embryos having spindles with focused poles

centrosomes are unstable and tend to divide or form other MTOC during anaphase even if spindles have focused poles during metaphase. These phenomena were not previously reported in other mammalian zygotes and may be specific to humans. To clarify the details, many human zygotes and further experiments are necessary. Figure 6 shows the multinucleation model. Our study showed that there was heterogeneity in the first mitotic spindle, which was strongly associated with multinucleation. High-AR spindles were stable and led to a mononuclear type. On the other hand, low-AR spindles often lead to multinucleation. In low-AR spindles, the poles often defocused during the last half of metaphase. Many multinuclei are modified to form mononuclei after the second mitosis since pole defocusing was firmly reduced. Thus, spindle stability is important for nuclear formation. Further investigations are necessary to uncover the

**Fig. 3 | Spindle pole defocusing causes multinucleation. a** Time-lapse live imaging of representative confocal microscopic images of a human embryo exhibiting a spindle with a unilateral defocused pole. The white arrow indicates the spindle pole. The unilateral pole (not attached to the sperm tail) changed during the last half of the metaphase. The blue arrowheads indicate the sperm tail. Scale bar, 10 μm. **b** Time-lapse live imaging of representative confocal microscopic images of a human embryo that has a spindle with a bilateral defocused pole. The white arrow indicates the spindle pole. Note that the spindle has bilateral focused poles, and a sperm tail is attached at the pole when spindle formation is completed(ii); however, the bilateral poles are defocused, and the sperm tail moves to the edge of the defocused pole from the center during the last half of metaphase(iii). Scale bar, 10 μm. The blue arrowheads indicate the sperm tail. White asterisks indicate cytoplasmic granules labeled with SPY505-DNA and SPY650-Tubulin. **c** Aspect ratio (AR) quantification of all imaged embryo spindles undergoing bipolar segregation and with focused or defocused poles. Box and whisker plots represent minimum, lower quartile, median, upper quartile, and maximum. Outliers fall beyond these lines. The bottom of the box is the first quartile (25th percentile) and the top of the box is the third quartile (75th percentile). The line in the middle of the box is the median

(50th percentile). A two-tailed Mann–Whitney U-test was performed. Source data are provided as a Source data file. **d** Quantification of spindle pole defocusing in mononuclear type (first bar) and multinuclear type (second bar) at 2-cell stage embryos undergoing bipolar segregation. Numbers are shown as bars. P-value from two-sided Fisher's exact test. Source data are provided as a Source data file. **e** Quantification of spindles with focused or defocused poles in imaged embryos undergoing bipolar segregation. As defocused poles can be unilateral or bilateral, this is detailed for the defocused pole in the second bar. In addition, unilaterally defocused poles could be of the pole-attached sperm tail or not a pole-attached type; detailed data are shown in the third bar. Numbers are shown as bars. **f** Aspect ratio (AR) quantification of all imaged embryo spindles undergoing bipolar segregation and with focused poles in multinuclear and mononuclear types at the two-cell stage. Box and whisker plots represent minimum, lower quartile, median, upper quartile, and maximum. The bottom of the box is the first quartile (25th percentile) and the top of the box is the third quartile (75th percentile). The line in the middle of the box is the median (50th percentile). An unpaired two-tailed Student's t-test was performed. Source data are provided as a Source data file.

mechanism underlying these phenomena. Revealing the mechanism underlying spindle formation in cleavage-stage embryos will contribute to improving the success rate of assisted reproduction technologies in humans.

## Methods

### Ethical approval
This study was approved by the Ethics Committee of Akita University (approval number: 1090-2), Japan Society of Obstetrics and Gynecology (Registry No. 75), and Asada Ladies Clinic (approval number: 2022-13). Participation in this study was entirely voluntary and no financial inducements were given for embryo donation. We obtained written informed consent from patients who donated human embryos. Donated embryos were determined as "Not Human Subjects Research." The patients were informed of the conditions of the donation, objectives, and methodology of human embryo research. They were offered counselling and alternative options, including discarding embryos and continued cryopreservation. Embryo donors were also informed that their donation would not affect their IVF cycle. The experiments were performed in accordance with the relevant regulations on human sperm/ovum/fertilized eggs set forth by the Japan Society of Obstetrics and Gynecology.

### Human embryos
Frozen human two-pronuclear stage embryos were donated for research by couples who had completed fertility treatment. The mean age of patients at embryo freezing was $32.00 \pm 3.03$ (SD) years. All surplus embryos were fertilized by IVF or ICSI and frozen on day 1, $19.7\,h \pm 0.92\,h$ (SD) after insemination. All samples were de-identified after freezing. All human embryos were thawed using Cryotop Safety Kit (Kitazato Corporation, Shizuoka, Japan) according to the manufacturer's protocol and cultured for 30 min in HiGROW OVIT Plus (Fuso Pharmaceutical Industries, Ltd., Osaka, Japan). Medium was covered with mineral oil in an incubator at 37 °C, with 5% $O_2$, 5% $CO_2$, and 90% $N_2$.

### Reagents for the detection of DNA and tubulin
Thirty-one human embryos were cultured in CultureCoin (Esco Medical, Egaa, Denmark) containing the live-cell stain probes SPY505-DNA (1:1000, Spirochrome, Thurgau, Switzerland) and SPY650-Tubulin (1:2000, Spirochrome) diluted in prewarmed HiGROW OVIT Plus (Fuso Pharmaceutical Industries) in a time-lapse incubator (Miri® Time-Lapse incubator; Esco Medical) at 37 °C, with 5% $O_2$, 6% $CO_2$, and 89% $N_2$. The embryos were captured every 5 min for at least 2 h to ensure that they were well dyed.

### Live imaging
Thirty-one human embryos were imaged during the first mitosis. After the cytoplasmic halo in human zygotes disappears, PNBD occurs[41]. Thus, immediately after the cytoplasmic halo started to disappear, the embryos were transferred to a 20-μL drop of observation culture medium that was placed in a 35-mm glass-bottom dish (Matsunami Glass Industry, Osaka, Japan) under mineral oil (Supplementary Fig. 1). Human embryos were imaged using an LSM980 laser scanning confocal microscope (Zeiss Japan, Tokyo, Japan) equipped with an objective C-Apochromat 40x/NA 1.20 W Korr and controlled by the ZEN Blue software (Zeiss). Images were acquired at a resolution of $0.414 \times 0.414$ μm per pixel, every 10–20 min ($60 \times 1.25$ μm optical sections) for a period of $8.4 \pm 3.1$ (SD) h and a depth of 16 bits. Fluorescent images were acquired using 488 and 639 nm lasers and brightfield images. Imaging was performed in an incubation chamber (Tokai Hit, Shizuoka, Japan) with the following parameters: temperature, 37 °C, with 5% $O_2$, 6% $CO_2$, and 89% $N_2$. After live-cell imaging, the embryos were placed in CultureCoin® containing the stain again. At least 3 h after the embryo reached the 4-cell stage, it was imaged again using the same microscope, not continuously but only once (Supplementary Fig. 1).

nine embryo was imaged in the second mitotic cycle. The embryos were cultured in CultureCoin until they reached the 2-cell stage. Images were acquired every 10–20 min ($80 \times 1.25$ μm optical sections) for a 5.0 h (IQR = 5.7 h) period. The other conditions and imaging settings were the same as those used for imaging performed during the first mitosis.

### Time-lapse imaging
Sixty-two human pronuclear stage embryos were transferred into CultureCoin® in a time-lapse incubator (Miri® Time-Lapse incubator, Esco Medical) after thawing. The wells containing 22 embryos (control group) were filled with 25 μL of HIGROW OVIT per embryo, and those containing 40 embryos (dyed group) were filled with 25 μL of the live-cell stain probes (at the same concentration used for imaging the embryos under mineral oil; Supplementary Fig. 1). The videos of the embryos were captured every 5 min during the time-lapse period. All videos were analyzed with the Miri® TL Viewer software (Esco Medical).

### Image analysis
three-dimensional (3D) visualizations of human embryos were performed using the Imaris software (version 10.0; Oxford Instruments). A median filter was applied prior to analysis. Spindle images for analysis were captured in TIFF format ($500 \times 500$ pixels) with

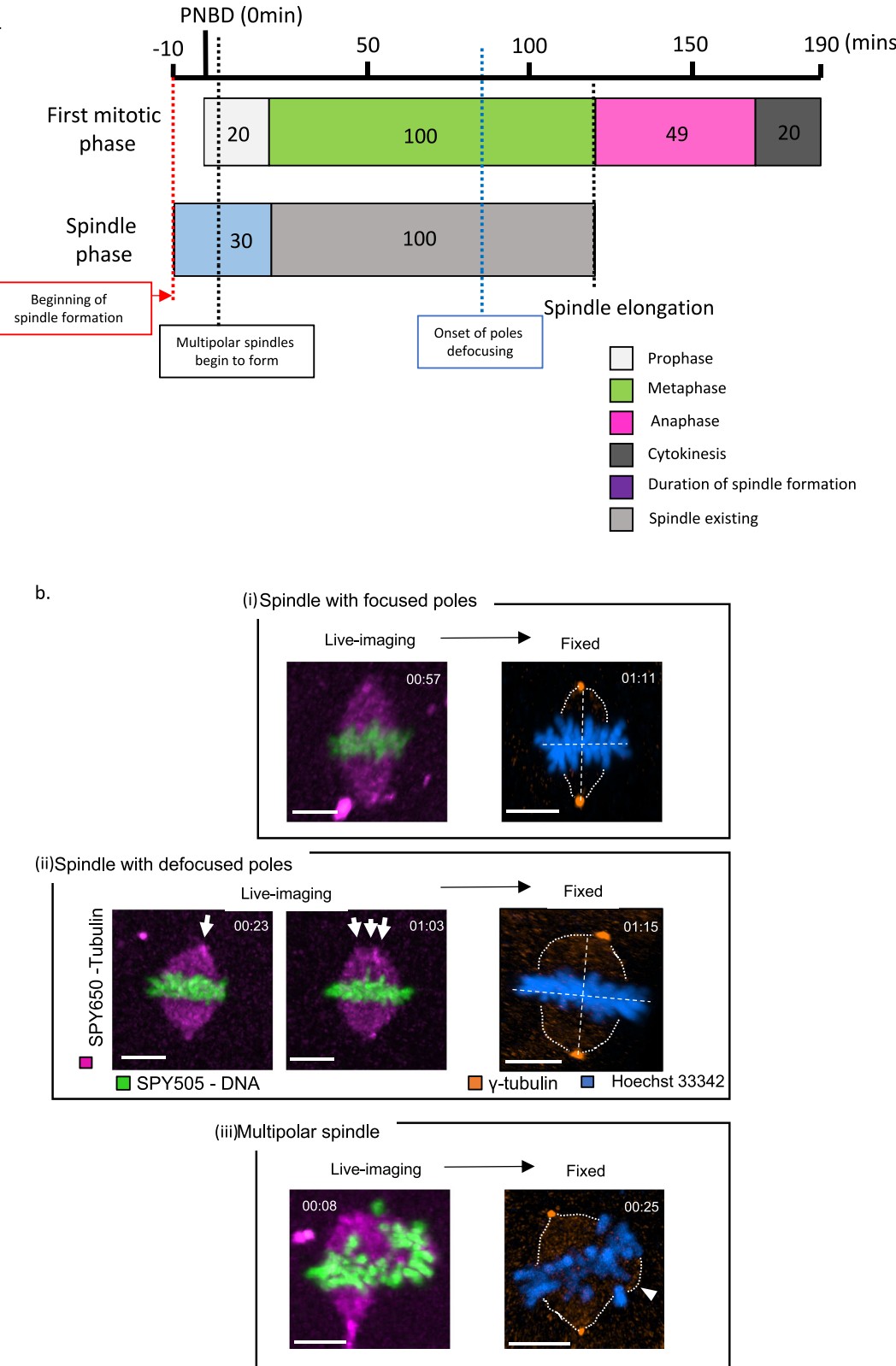

**Fig. 4 | Multipolar spindles and defocused spindles do not have two focused poles and cause multinucleation, although the formation processes differ.**
**a** Median durations of each mitotic phase and spindle phase plotted from 31 imaged embryos during the first mitosis. The black dashed line indicates median times for beginning multipolar spindle formation. The blue dashed line indicates median times for pole defocusing. Source data are provided as a Source data file. **b** Live-imaging and immunofluorescence staining of γ-tubulin and chromosomes (Hoechst33342) in human zygotes. Scale bar, 10 mm. The white arrow indicates the spindle pole. The white arrowhead indicates acentriolar microtubule organizing centers (aMTOC).

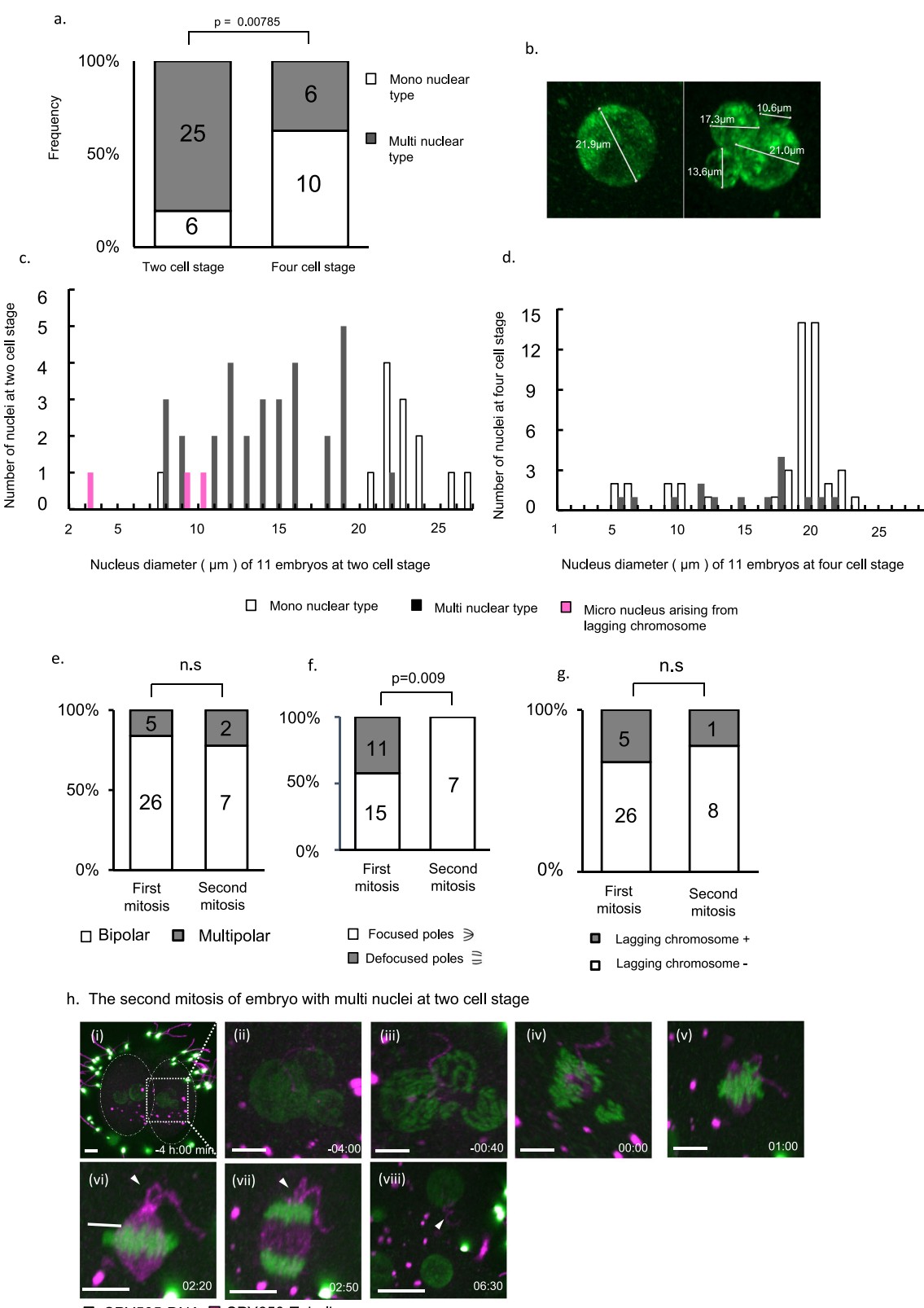

h. The second mitosis of embryo with multi nuclei at two cell stage

SPY505-DNA   SPY650-Tubulin

500% zoom using a Snapshot in Imaris10.0 (Oxford Instruments). This image was captured at the end of metaphase from an angle where the chromosomes appeared to be aligned in a straight line and where the spindle appeared to have the maximum area. Each captured image was processed using the Pixel Classification algorithm embedded in Ilastik version 1.4.0, image recognition software[42,43] and only the shape of the spindle was extracted from the background as a region of interest. Each processed image was binarized, and the aspect ratio, a ratio of major and minor axis, of each image was calculated automatically using Fiji[44]. This "aspect ratio" is one of the basic amounts of image characteristics; the range is >1, with 1 meaning close to a true circle.

The morphology of the nuclei was analyzed at least 3 h after the first or second division. Only spherical nuclei were measured, and

**Fig. 5 | Stable second mitotic spindles modify multinuclei to normal nuclei. a** Quantification of nucleation status in all imaged embryos at the 2-cell stage. The second bar shows the quantification of nuclear errors at the 4-cell stage and the p-value from a two-sided Fisher's exact test. Source data are provided as a Source Data file. **b** Diameter of each nucleus was measured at the 2- and 4-cell stages. **c, d** Histogram showing nuclear diameter distribution of imaged embryo at 2- and 4-cell stages grouped into 1-μm bins. (**c**) and (**d**) show different stages of the same embryos (*n* = 11). Source data are provided as a Source data file. **e** Quantifying anaphase errors in all imaged embryos during the first mitosis. The second bar shows the quantification of anaphase errors during the second mitosis and the *p*-value from a two-sided Fisher's exact test. *p*-value = 0.645. Source data are provided

as a Source data file. **f** Quantifying spindle pole defocusing in bipolar spindles during the first (first bar) and the second (second bar) mitosis. Numbers are shown as bars. *P*-value from two-sided Fisher's exact test. Source data are provided as a Source data file. **g** Quantifying the lagging chromosome in all imaged embryos during the first (first bar) and the second (second bar) mitosis. Numbers are shown as bars. *P*-value from two-sided Fisher's exact test. *p*-value = 1. Source data are provided as a Source data file. **h** Time-lapse live imaging of representative confocal microscopy images of the second mitosis in human embryos with multiple nuclei at the 2-cell stage. The white arrowhead indicates the sperm tail. White arrows indicate sperms. 3D Scale bar: 10 μm.

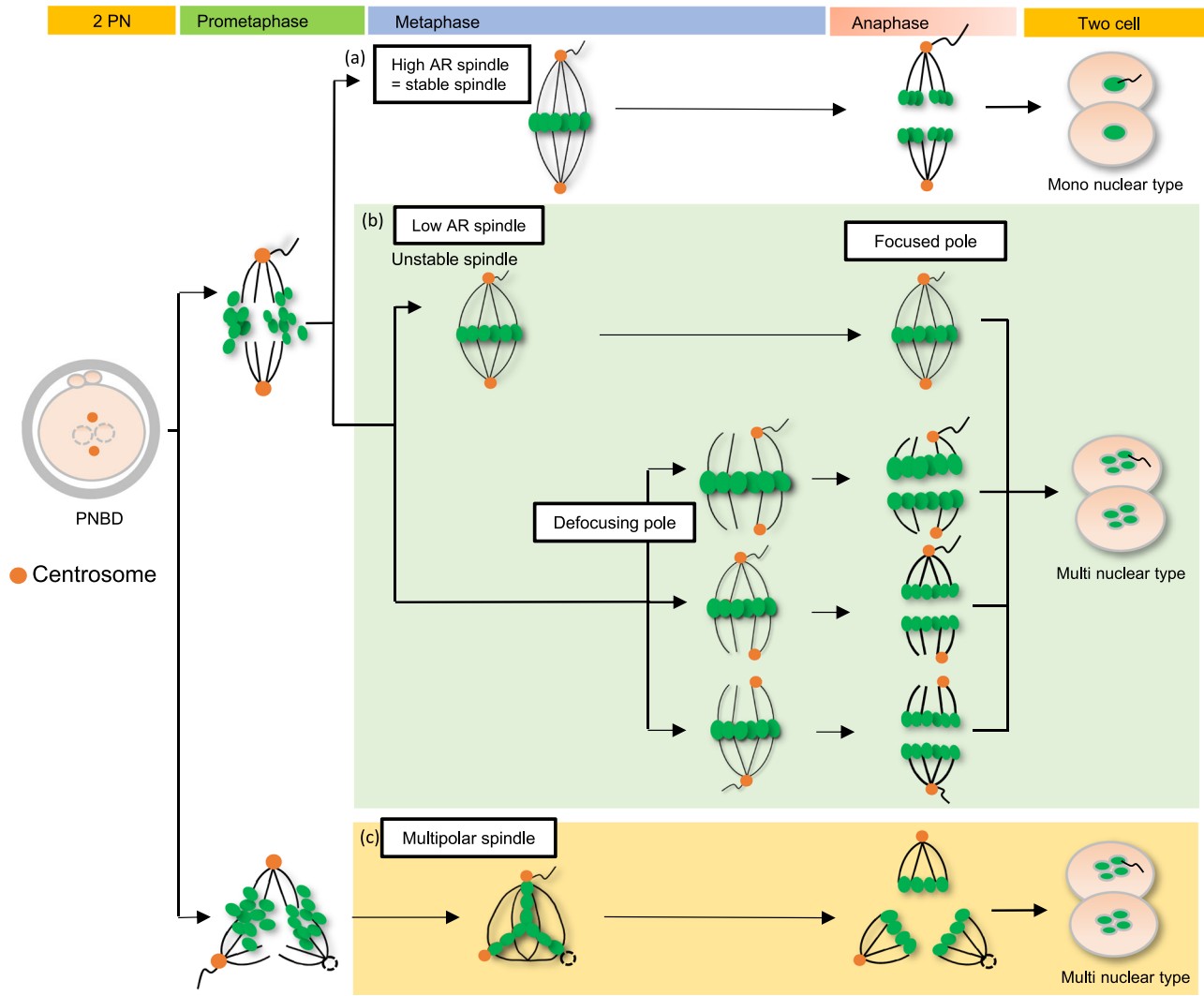

**Fig. 6 | Model of how spindle instability causes multinucleation during the first mitosis.** After the PNBD, the spindle formation begins at the poles. There were two pole-number patterns: two and three poles. In bipolar spindles, high-aspect-ratio (AR) spindles are stable (**a**), have focused poles, and lead to mononuclei. However, in bipolar spindles, low-AR spindles are unstable (**b**), and centrosomes inherited

from the sperm cannot maintain microtubule nucleation centers. Consequently, these spindle poles often become defocused, leading to multiple nuclei. Although multipolar spindles (**c**) have three poles, they are stable and do not change until the anaphase begins. As a result, all multipolar spindles lead to multinuclei.

nuclei that were difficult to recognize and did not exhibit spherical shapes were not measured.

**Immunofluorescence after live imaging**
Four human 2PN zygotes were live-imaged using the same method. After PNBD, the zygotes were imaged for 30.0 ± 21.6 [SD]mins and fixed after 66.5 ± 29.6 [SD] mins from PNBD.

Human 2PN zygotes were fixed for 60 min at 37 °C in 100 mM HEPES (pH 7, titrated with KOH), 50 mM EGTA (pH 7, titrated with KOH), 10 mM MgSO$_4$, 2% formaldehyde (MeOH free) and 0.2% Triton X-100, based on previously published methods[37]. Human zygotes were left in PBS containing 0.1% Triton X-100 overnight at 4 °C. All antibody incubations were performed in PBS, 3% BSA, and 0.1% Triton X-100, either overnight at 4 °C (for primary antibodies) or for 2 h

at room temperature (for secondary antibodies). The primary antibody was mouse anti-γ-tubulin (1:500, GTU88, T5326, lot number:0000216733, Sigma-Aldrich). The secondary antibody was Alexa-Fluor-555 labeled anti-mouse (1:500, ab150118, lot number:1062774-2, Abcam). DNA was stained with 0.5 µg/ml Hoechst 33342 (Molecular Probes). Samples were imaged using LSM980 laser scanning confocal equipped with an Airyscan detector (Zeiss, Japan) using equipped with an objective C-Apochromat 40x/NA 1.20 W Korr.

### Statistical analysis

All statistical analyses were performed using EZR[45], a modified version of R, with statistical functions frequently used in biostatistics. An unpaired two-tailed Student's *t*-test was used for two groups (Figs. 2c, d and 3f) after assessing their distribution with the Kolmogorov-Smirnoff test. Two-tailed Mann–Whitney U-test (Fig. 3c, S5d), or a two-sided Fisher's exact test (Figs. 1g, 3d, 4a, 4e–g, S5b, S5c), and Kruskal-Wallis tests were used to analyze more than two groups (Fig. 1c).

### Reporting summary

Further information on research design is available in the Nature Portfolio Reporting Summary linked to this article.

## Data availability

The raw data from this study are available from the corresponding author upon request. The primary confocal microscopy data were not uploaded to a data repository because of their large size but are available from the authors upon request. Source data are provided with this paper.

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

## Acknowledgements

We gratefully acknowledge Akihiro Ichikawa of Carl Zeiss Co. Ltd. and Soichi Koyota of Molecular Medicine Laboratory, Bioscience Education and Research Support Center, Akita University, for their support and assistance with image analysis in this work. We also thank Kota Saito and Miharu Maeda of Department of Biological Informatics and Experimental Therapeutics, Graduate School of Medicine, Akita University for advice about cell incubation. We would also like to thank Editage (www.editage. com) for their English language editing services. Y.O. is supported by the Imai Foundation. Y.T. is supported by the JSPS KAKENHI (https://www. jsps.go.jp/english/index.html; grant numbers 21H03073 and 21K19549).

## Author contributions

Y.O., K.T., and Y.T. designed the study. Y.O. and M.G. conducted the experiments. N.F. and Y.A. collected the human embryos and obtained ethical approval for the study. Y.O. performed the data analysis and interpretation. Particle analysis was performed by T.O.; Y.O. wrote the manuscript. The method of particle analysis was developed by T.O., H.S., K.T., M.O., Taichi Sakaguchi, T.H., T.I., A.F., Tae Sugawara, K.M., H.M., Y.K., and Y.T. All authors reviewed the final manuscript.

## Competing interests

The authors declare no competing interests.
