## [Peer Review File · Nature Communications]

Shape of the first mitotic spindles impacts multinucleation in human embryosREVIEWER COMMENTS

Reviewer #1 (Remarks to the Author):

This paper from Ono et al performs dual colour live embryo imaging of the human zygote progressing through the first embryonic cleavage division. This study represents the first live imaging of the human mitotic spindle during the zygotic cleavage division. This division is of great interest because, 1) it is the first time the two parental genomes unify and segregate on the same mitotic spindle, and 2) there is growing evidence that this first mitotic division is error prone, with Day 3 embryos already carrying a high frequency of mitotic aneuploidies. Thus, chromosome missegregation during the early mitotic divisions is a major route to chromosomal mosaicism - a hallmark of human embryos. Another feature in early embryos is the presence of multinucleated blastomeres at the 2-cell stage. The origin of these is unclear although multipolar spindles may explain formation of multiple cells during these divisions. Understanding the spindle dynamics and mechanisms underlying these events is a cutting edge area of human reproduction research.

At the heart of this study is data showing how the mitotic spindle poles become defocused during metaphase of the first zygotic mitosis. The authors propose that this causes multinucleation at the 2-cell stage. The live imaging is fantastic: visualisation of a first mitosis showing creation of a multinucleated blastomere from a bipolar spindle is a key observation. The generation of a mononucleated cell from a multinucleated cell is also significant because the origin of multiple DNA masses when the spindle is bipolar was mysterious. The authors go on to propose that this spindle pole instability is due to either diminishing KIFC1 (a spindle pole focusing motor), or inherent dysfunction of the sperm centrosome. These are nice ideas but the data to support them is not yet in place. Moreover, there are some control experiments and further analysis that are needed to substantiate the key findings mentioned above. Other experiments are then presented, but these look to be either extraneous (separation of maternal and paternal genomes) to the main thread of the paper, or not particularly novel (see full comments below). However, I do think there are important observations in here and that with additional experiments it could make an important contribution to the field.

Major Points

1. The introduction does not accurately reflect the status of the field, where live imaging of structures within human embryos has already been established, including human zygotes during the first mitotic division (Domingo-Muelas et al, 2023 doi: 10.1016/j.cell.2023.06.003, Currie et al, 2022, <https://doi.org/10.1038/s41467-022-34294-6>, Cavazza et al, 2021 doi: 10.1016/j.cell.2021.04.013). What has not been established is dual colour live imaging of the spindle and chromosomes, which is where this paper has novelty. Authors should also cite primary literature in the introduction rather than reviews (especially when the cell biology of human mitosis 1 is covered by only a few papers).
2. The methods state that the material used was frozen on day 2 post-fertilisation at the 2PN stage, suggesting these zygotes are severely delayed (they should be at cleavage stage at this point). Please can authors clarify their rationale for using this material and why they are confident that conclusions on human embryo development can be reached.
3. In Figure 1 the authors must report the frequency of multinucleation in their control, imaged and dyed embryo groups. This is important as authors state that their experiments show much higher levels of multinucleation than others (line 225-231). I am concerned that this could be due to phototoxicity arising from the dual colour time-lapse microscopy. Authors should be able to provide this data from existing time lapse imaging movies presented in Figure 1c.
4. I was left wondering about the classification of multipolar spindle (Fig. 1B) and bipolar spindles with defocused poles (Fig. 3a). Is a multipolar division just an extreme case of pole defocusing, or are they fundamentally different i.e. wrong number of centrosomes? If so how is that possible? Authors need to clarify the differences in their classification systems and check the location/number of centrosomes in these different spindle types. This is also relevant to how the model is presented in Figure 6.
5. In Figure 3 the authors conclude that pole defocusing is the cause of multinucleation. This experiment (Fig. 3d) shows that mononuclear outcomes never arise when there is pole defocusing. However, focused poles can give rise to multi-nucleation in 9/20 embryos. This is close to 50/50.

Are the authors sure that there is no defocusing in the z-axis?. If not then there must be at least one other route to multinucleation, when poles remain focused. Here is where further experimentation is needed. Can the authors show that re-focusing the poles rescues all the observed multinucleation phenotype or only a subset? Based on reading the discussion it looks like the authors suggest the mechanism may involved KIFC1. The authors should inject mRNA and show a rescue. This has been done in Meiosis I so is feasible. This will also show whether the multinucleation from focused spindles is a result of the same, or different mechanism.

6. The authors also suggest in the discussion that, "dysfunction of the centrosome inherited from the sperm..." as contributory factor. The description of this is somewhat unclear: I think the authors are proposing that the pole with the sperm tail inhibits defocusing leaving a unilateral phenotype (Fig.3e). The writing needs improvement here. The authors should then characterise the centrosomes in these spindles using immunofluorescence and provide some evidence to support this (nice) idea. i.e. look at centriole number, amount of pericentriolar material etc etc

7. The data in Figure 4 quantifying the shift from multinucleated blastomeres at the 2-Cell stage, to mononucleated at later stages is not novel. The authors state this themselves (lines 297-300). What is not known is whether the spindle fragmentation reported in the 2-cell stage is also reduced at later stages. This would be expected if the authors model is correct and spindle defocusing is a primary cause of multinucleation. The authors should carry out further live imaging of the second mitosis and quantify this (and behaviour of sperm tail) – as per Fig 3. I think they only have one mitosis 2 currently so the n needs increasing to reach any meaningful results.

8. There appear to be micronuclei in Fig 4e? Are these also result of defocusing or due to presence of lagging chromosomes as observed in human and bovine embryos?

9. The authors go on to provide evidence (in Fig 5) on how the two parental genomes are partitioned in the first zygotic mitosis. This analysis does not fit with the story at all. Also, the tracking of parental genomes via post-hoc, hypothetical labelling of chromosomes does not look robust. We suggest authors remove this data and the associated discussion. Otherwise, the authors must directly label the parental genomes in different fluorophores (for example) and then track their progression through the first mitosis. We appreciate the methodological and ethical challenges in performing such an experiment with human zygotes, however the conclusions drawn from the current analysis presented do not rigorously test the hypothesis that parental genomes remain separated.

10. This paper needs some rewriting as it is difficult to follow in many places. The discussion is far too long and even the title does not really make sense, "...impacts nucleation" – I would suggest they reword to mention spindle pole defocusing can lead to multinucleation at the 2-cell stage (main result of paper). There are also several inaccuracies too: line 50 is incorrect as fixed studies do not provide information on the dynamics of the cytoskeleton or chromosomes. Line 36: many "...errors, such as missegregation ..." – need to say chromosome missegregation.

Minor points

1. Figure 2b iii – provide the same labelling of aspect ratio as above.
2. Images are not clear in 3B panel 5, clarify the outcome for nucleation status.
3. Clarify the terminology around control, dyed and imaged in the main text and figure 1c.
4. Define direct cleavage in Figure 1e (is it into 3 cells?)
5. In the abstract language needs to be around defocusing, not AR of the spindle. This needs to be consistent throughout the paper as this is confusing.
6. Could the authors provide the data showing the time at which defocusing occurs; to what extent does this vary between embryos?

Reviewer #2 (Remarks to the Author):

The manuscript NCOMMS-24-03614-T entitled "Shape of the first mitotic spindles impacts nucleation in human embryos" aims to visualize spindle formation in human zygotes with an innovative approach to increase our understanding of a critical step for embryo development. In the last 10 years, many efforts were made mostly to understand the origin of chromosome segregation errors during oocyte meiosis in humans, highlighting the presence of unstable spindles

with unfocused spindle poles. However, understanding the causes of aneuploidy in human embryos has been more challenging.

Here the authors made a tremendous effort to visualize for the first time spindle and chromosome behavior during the first two rounds of mitosis in human embryos. They use an innovative approach using spy-dyes that allows to visualization of different structures without the need to manipulate the levels of expression of different components of the cell. Similar to previous studies they observed that first mitosis in human embryos is error-prone. However, they add a new layer to this analysis showing that those errors were the product of the formation of multipolar spindles or spindles that become unstable with unfocused poles during metaphase. Altogether, this study provides important information about how the first couple of mitotic division occurs in human embryos, opening the door to new hypotheses about the mechanisms of spindle formation in early embryos.

I have some comments for the authors:

1. I was wondering if those embryos where they see abnormal spindles with multinucleated blastomeres can reach to blastocyst stage. I know that it would not be feasible to image during that long time, but after the imaging, if those embryos were back in the incubator, can continue their development to blastocyst?
2. The authors were able to get highly meaningful data about the spindle morphology and chromosome behavior, but I consider they can extract more information from those time lapses: For example The timing of entering to anaphase change among embryos with different spindle morphology, or between spindle with focused or unfocused poles? Are there changes in timing associated with the result of multinucleation? The defocusing of the spindle poles is happening always at the same time in different embryos. This type of information can help the field to start to understand the mechanism for these errors.
3. I strongly suggest that the authors discuss their results in light of the data published a couple of years ago in Currie et al 2022.
4. Finally, I do not see a clear connection between all the analyses on spindle building and multinucleation and the last part about parental chromosome separation. Is there any change in the parental genome distribution with the different types of spindle morphologies? What is the rationale for doing this analysis? Could the authors clarify this point?

Minors

5. Figure 1. I think that panel f should be panel e to be better associated with the images in panel d

REVIEWER COMMENTS

Reviewer #1 (Remarks to the Author):

This paper from Ono et al performs dual colour live embryo imaging of the human zygote progressing through the first embryonic cleavage division. This study represents the first live imaging of the human mitotic spindle during the zygotic cleavage division. This division is of great interest because, 1) it is the first time the two parental genomes unify and segregate on the same mitotic spindle, and 2) there is growing evidence that this first mitotic division is error prone, with Day 3 embryos already carrying a high frequency of mitotic aneuploidies. Thus, chromosome missegregation during the early mitotic divisions is a major route to chromosomal mosaicism - a hallmark of human embryos. Another feature in early embryos is the presence of multinucleated blastomeres at the 2-cell stage. The origin of these is unclear although multipolar spindles may explain formation of multiple cells during these divisions. Understanding the spindle dynamics and mechanisms underlying these events is a cutting edge area of human reproduction research.

At the heart of this study is data showing how the mitotic spindle poles become defocused during metaphase of the first zygotic mitosis. The authors propose that this causes multinucleation at the 2-cell stage. The live imaging is fantastic: visualisation of a first mitosis showing creation of a multinucleated blastomere from a bipolar spindle is a key observation. The generation of a mono-nucleated cell from a multinucleated cell is also significant because the origin of multiple DNA masses when the spindle is bipolar was mysterious. The authors go on to propose that this spindle pole instability is due to either diminishing KIFC1 (a spindle pole focusing motor), or inherent dysfunction of the sperm centrosome. These are nice ideas but the data to support them is not yet in place. Moreover, there are some control experiments and further analysis that are needed to substantiate the key findings mentioned above. Other experiments are then presented, but these look to be either extraneous (separation of maternal and paternal genomes) to the main thread of the paper, or not particularly novel (see full comments below). However, I do think there are important observations in here and that with additional experiments it could make an important contribution to the field.

Our study shows that the cause of multinucleation at the two-cell stage in human embryos is the shape of the spindle. In other words, low-aspect ratio (AR)

spindles with defocused poles and low-AR with focused poles and multipolar spindles also cause multinucleation. Indeed, spindles with defocused poles always undergo multi nuclei and low-AR spindles more often have poles defocusing during the last half of metaphase than high-AR spindle, although low-AR spindles with focused poles also tend to undergo multi nuclei (Fig. 3f). Therefore, we state that the shape of the spindle is firmly associated with multinucleation in this paper.

We deleted the section about separation of maternal and paternal genomes.

To ensure our hypothesis, we performed additional experiments to investigate how sperm centrosomes attached on the first abnormal spindles and shapes of the second spindle.

Major Points

1. The introduction does not accurately reflect the status of the field, where live imaging of structures within human embryos has already been established, including human zygotes during the first mitotic division (Domingo-Muelas et al, 2023 doi: 10.1016/j.cell.2023.06.003, Currie et al, 2022, <https://doi.org/10.1038/s41467-022-34294-6>, Cavazza et al, 2021 doi: 10.1016/j.cell.2021.04.013). What has not been established is dual colour live imaging of the spindle and chromosomes, which is where this paper has novelty. Authors should also cite primary literature in the introduction rather than reviews (especially when the cell biology of human mitosis 1 is covered by only a few papers).

We thank the reviewers for these references and have included them in the revised text. Additionally, we have replaced the references in the introduction (line 78-86).

2. The methods state that the material used was frozen on day 2 post-fertilisation at the 2PN stage, suggesting these zygotes are severely delayed (they should be at cleavage stage at this point). Please can authors clarify their rationale for using this material and why they are confident that conclusions on human embryo development can be reached.

We apologize for the incorrect sentence "2PN stage, 2nd day after fertilization". The correct sentence is Day 1.

We used human 2PN zygotes frozen 19 hours after insemination. We revised this section (line 616-617).

3. In Figure 1 the authors must report the frequency of multinucleation in their control, imaged and dyed embryo groups. This is important as authors state that their experiments show much higher levels of multinucleation than others (line 225-231). I am concerned that this could be due to phototoxicity arising from the dual colour time-lapse microscopy. Authors should be able to provide this data from existing time lapse imaging movies presented in Figure 1c.

We have added data on the frequency of multinucleation in the control and dyed embryo groups (Fig 1g, line 143-148). The frequency of multinucleation in the control (75%) and dyed group (79%) was not significantly different. The frequency in the imaged group was 81%. We think the frequency of the imaged group is more accurate and tends to higher frequency than the other group because the embryos in the imaged group were stained and evaluated by a confocal microscopy and it was easy to find multi nuclei. Therefore, we do not think phototoxicity severely affected the occurrence of multinucleation.

4. I was left wondering about the classification of multipolar spindle (Fig. 1B) and bipolar spindles with defocused poles (Fig. 3a). Is a multipolar division just an extreme case of pole defocusing, or are they fundamentally different i.e. wrong number of centrosomes? If so how is that possible? Authors need to clarify the differences in their classification systems and check the location/number of centrosomes in these different spindle types. This is also relevant to how the model is presented in Figure 6.

We think bipolar spindles with defocused poles and multipolar spindle are fundamentally different. Bipolar spindles with defocused poles have focused two poles in the beginning of metaphase but changed the defocused poles in the last half of metaphase. In contrast, multipolar spindles had three poles immediately after pronuclear envelope breakdown. Moreover, we performed additional experiments to investigate the number of centrosomes in each spindle and their positions. We visualized γ -tubulin in human zygotes using live-imaging and fixed methods and proceeded our study We thank you for the nice advice about the centrosomes. The results are shown in Fig 4c. High-aspect ratio spindle with focused poles had two γ -tubulin positive poles, and the two poles were lined up on a line

perpendicular to the chromosomes (Fig. 4b(i)). In contrast, spindles with defocused poles had two γ -tubulin positive poles, but the position was offset from the line perpendicular to the chromosomes (Fig. 4b(ii)). Multipolar spindle appeared to have three poles, but actually only have two γ -tubulin positive poles and the other pole was γ -tubulin negative (Fig. 4b(iii)). Therefore, spindles with defocused poles are significantly different from multipolar spindles owing to the formation process and the location of the centrosome. We added Fig. 4b and described how different these spindles are formed. Additionally, we revised the model in Fig 6.

5. In Figure 3 the authors conclude that pole defocusing is the cause of multinucleation. This experiment (Fig. 3d) shows that mononuclear outcomes never arise when there is pole defocusing. However, focused poles can give rise to multi-nucleation in 9/20 embryos. This is close to 50/50. Are the authors sure that there is no defocusing in the z-axis?. If not then there must be at least one other route to multinucleation, when poles remain focused. Here is where further experimentation is needed. Can the authors show that re-focusing the poles rescues all the observed multinucleation phenotype or only a subset? Based on reading the discussion it looks like the authors suggest the mechanism may involved KIFC1. The authors should inject mRNA and show a rescue. This has been done in Meiosis I so is feasible. This will also show whether the multinucleation from focused spindles is a result of the same, or different mechanism.

Our study shows that low-AR spindles (including those with defocused and focused poles) cause multinucleation during the first mitosis in human embryos, showing that not only defocusing poles induce multinucleation in bipolar spindles. We ensured that there is no defocusing in focusing poles by checking the 3-dimensional images.

However, we think this is an excellent suggestion about microinjecting into human embryos to investigate if KIFC1 can rescue multinucleation. However, we need to get another patient cohort to perform this new experiment, as we have had to seek additional ethical approvals to allow patient consent for microinjection into human embryos. This experiment is well beyond the scope of the current study.

In our study, over three γ -tubulin spots were observed during anaphase (Fig.S6). This suggested that zygotic centrosomes are unstable, and tend to divide or form other MTOC during anaphase even if spindles have focused poles during metaphase. This phenomenon may explain why low-AR spindle with focused poles tend to be multinucleated.

6. The authors also suggest in the discussion that, “dysfunction of the centrosome inherited from the sperm...” as contributory factor. The description of this is somewhat unclear: I think the authors are proposing that the pole with the sperm tail inhibits defocusing leaving a unilateral phenotype (Fig.3e). The writing needs improvement here. The authors should then characterise the centrosomes in these spindles using immunofluorescence and provide some evidence to support this (nice) idea. i.e. look at centriole number, amount of pericentriolar material etc etc

We thank the reviewer's suggestion. We performed the additional experiments using immunofluorescence.

We used antibody for gamma-tubulin which is a centrosomal component involved in microtubule nucleation to investigate the number of centrosomes in each spindle and to determine their positions. The results were shown in Fig 4b. We discovered that the centrosomes have different positions. Spindles with defocused poles had two γ -tubulin positive poles, but the position was offset from the line perpendicular to the chromosomes (Fig.4c(ii)). In contrast, the spindle with focused poles had two γ -tubulin positive poles, and the two poles lined up on a line perpendicular to the chromosomes (Fig.4b(i)). We expect that sperm centrosomes were unstable and could not maintain the function as microtubule organizing centers during longer metaphase and cause pole defocusing.

7. The data in Figure 4 quantifying the shift from multinucleated blastomeres at the 2-Cell stage, to mononucleated at later stages is not novel. The authors state this themselves (lines 297-300). What is not known is whether the spindle fragmentation reported in the 2-cell stage is also reduced at later stages. This would be expected if the authors model is correct and spindle defocusing is a primary cause of multinucleation. The authors should carry out further live imaging of the second mitosis and quantify this (and behaviour of sperm tail) – as per Fig 3. I think they only have one mitosis 2 currently so the n needs increasing to reach any meaningful results.

We thank the reviewer for this comment. We have performed additional experiment on

second mitosis.

We have revised the result in Fig. 5e-g. We discovered that the second mitotic spindles had fewer poles defocusing than those of the first mitosis, as expected.

8. There appear to be micronuclei in Fig 4e? Are these also result of defocusing or due to presence of lagging chromosomes as observed in human and bovine embryos?

Some of the nuclei under 10 μm arise from lagging chromosomes during the first mitosis (Fig 5c). We have added data on lagging chromosomes in Figs. 5c and S5. We imaged the embryos ($n = 11$) in this histogram consecutively during the first mitosis and imaged them again one time after the second mitosis; therefore, we do not know if there are micronuclei arising from lagging chromosomes during the second mitosis, as shown in Fig. 5d.

9. The authors go on to provide evidence (in Fig 5) on how the two parental genomes are partitioned in the first zygotic mitosis. This analysis does not fit with the story at all. Also, the tracking of parental genomes via post-hoc, hypothetical labelling of chromosomes does not look robust. We suggest authors remove this data and the associated discussion. Otherwise, the authors must directly label the parental genomes in different fluorophores (for example) and then track their progression through the first mitosis. We appreciate the methodological and ethical challenges in performing such an experiment with human zygotes, however the conclusions drawn from the current analysis presented do not rigorously test the hypothesis that parental genomes remain separated.

We have deleted the section on separation of parental genomes according to the reviewer's suggestions.

10. This paper needs some rewriting as it is difficult to follow in many places. The discussion is far too long and even the title does not really make sense, "...impacts nucleation" – I would suggest they reword to mention spindle pole defocusing can lead to multinucleation at the 2-cell stage (main result of paper). There are also several inaccuracies too: line 50 is incorrect as fixed studies do not provide information on the dynamics of the cytoskeleton or chromosomes. Line 36: many "...errors, such as missegregation ..." – need to say

chromosome missegregation.

We have revised the discussion to ensure that it is shorter and simpler, as the reviewer suggested. Spindle pole defocusing, low-aspect ratio spindles, and multipolar spindles firmly induced multinucleation. We have reconsidered the title and rewritten it to improve reader understanding.

Minor points

1. Figure 2b iii – provide the same labelling of aspect ratio as above.

We have made the necessary revision.

2. Images are not clear in 3B panel 5, clarify the outcome for nucleation status.

We have replaced with a clearer image in Figure 3B, panel 5.

3. Clarify the terminology around control, dyed and imaged in the main text and figure 1c.

We have revised the point in the main text and the legend of Figure 1c.

4. Define direct cleavage in Figure 1e (is it into 3 cells?)

We have revised the point on direct cleavage (line 134-135).

5. In the abstract language needs to be around defocusing, not AR of the spindle. This needs to be consistent throughout the paper as this is confusing.

We have written that spindles with defocused poles and low-AR spindles with focused poles can induce multinucleation.

6. Could the authors provide the data showing the time at which defocusing occurs; to what extent does this vary between embryos?

We have added Figure 4a and Table S1.2 that demonstrate mitotic duration by spindles.

Reviewer #2 (Remarks to the Author):

The manuscript NCOMMS-24-03614-T entitled "Shape of the first mitotic spindles impacts nucleation in human embryos" aims to visualize spindle formation in human zygotes with an innovative approach to increase our understanding of a critical step for embryo development. In the last 10 years, many efforts were made mostly to understand the origin of chromosome segregation errors during oocyte meiosis in humans, highlighting the presence of unstable spindles with unfocused spindle poles. However, understanding the causes of aneuploidy in human embryos has been more challenging.

Here the authors made a tremendous effort to visualize for the first time spindle and chromosome behavior during the first two rounds of mitosis in human embryos. They use an innovative approach using spy-dyes that allows to visualization of different structures without the need to manipulate the levels of expression of different components of the cell. Similar to previous studies they observed that first mitosis in human embryos is error-prone. However, they add a new layer to this analysis showing that those errors were the product of the formation of multipolar spindles or spindles that become unstable with unfocused poles during metaphase. Altogether, this study provides important information about how the first couple of mitotic division occurs in human embryos, opening the door to new hypotheses about the mechanisms of spindle formation in early embryos.

Thank you for your comments. We believe our study can contribute to uncovering mechanisms of the first error prone mitosis in human embryos.

I have some comments for the authors:

1. I was wondering if those embryos where they see abnormal spindles with multinucleated blastomeres can reach to blastocyst stage. I know that it would not be feasible to image during that long time, but after the imaging, if those embryos were back in the incubator, can continue their development to blastocyst?

In our study, some multinucleated embryos at the two-cell stage could reach the blastocyst stage after imaging.

The blastocyst rate was 28% (n = 7/25) in multinucleated embryos and 50% (n = 3/6) in mononucleated embryos.

2. The authors were able to get highly meaningful data about the spindle morphology and chromosome behavior, but I consider they can extract more information from those time lapses: For example The timing of entering to anaphase change among embryos with

different spindle morphology, or between spindle with focused or unfocused poles? Are there changes in timing associated with the result of multinucleation? The defocusing of the spindle poles is happening always at the same time in different embryos. This type of information can help the field to start to understand the mechanism for these errors.

We have added information on the timing of each type of spindle in Fig. 4a and Table S1-2.

3. I strongly suggest that the authors discuss their results in light of the data published a couple of years ago in Currie et al 2022.

We have revised the introduction and discussion to compare the paper you recommended (line 78-80, 373-376).

4. Finally, I do not see a clear connection between all the analyses on spindle building and multinucleation and the last part about parental chromosome separation. Is there any change in the parental genome distribution with the different types of spindle morphologies? What is the rationale for doing this analysis? Could the authors clarify this point?

We have deleted the section on separation of parental genomes according to the reviewer's comments.

Minors

5. Figure 1. I think that panel f should be panel e to be better associated with the images in panel d

We have revised this point per the reviewer's suggestion.

REVIEWERS' COMMENTS

Reviewer #1 (Remarks to the Author):

The authors have done an excellent job of considering and addressing my questions/comments.

minor

- + I think there is an error in supplementary table 1 (multipolar n=6 when should be n=5)?
- + Could the authors note in their introduction that transferring embryos with multinucleation can give rise to live birth (albeit reduced) i.e. it is not a bona fide deselection marker?

Reviewer #2 (Remarks to the Author):

The authors address all my comments. I consider that the manuscript has been improved and recommended for publication.